# Redox–Amino Acid Metabolic Crosstalk in Ovarian Cancer Stem Cells: Integrating Metabolic Reprogramming, Signaling, and the Tumor Microenvironment

**DOI:** 10.3390/antiox14121413

**Published:** 2025-11-27

**Authors:** Dan Liu, Huawei Yi, Cunjian Yi

**Affiliations:** 1Department of Obstetrics and Gynecology, The First Affiliated Hospital of Yangtze University, Yangtze University, Jingzhou 434023, China; 2Laboratory of Oncology, Center for Molecular Medicine, School of Basic Medicine, Yangtze University, Jingzhou 434000, China; 3Central Laboratory, The First Affiliated Hospital of Yangtze University, Jingzhou 434000, China; 4Hubei Provincial Clinical Research Center for Personalized Diagnosis and Treatment of Cancer, Jingzhou 434000, China

**Keywords:** ovarian cancer stem cells, redox homeostasis, amino acid metabolism, tumor microenvironment, ferroptosis

## Abstract

Ovarian cancer stem cells (OCSCs) possess stemness; differentiation capacity; and tolerance to oxidative, metabolic, and therapeutic stress, driving recurrence and chemoresistance. Emerging evidence highlights a synergistic interplay between redox homeostasis and amino acid metabolism in maintaining stemness and treatment resistance. This review integrates redox regulation, amino acid metabolic reprogramming, and tumor microenvironment (TME) signals into a unified “redox–amino acid–TME” framework. OCSCs balance signal transduction and antioxidant defense by fine-tuning reactive oxygen species (ROS) levels. Glutamine, serine/glycine, and sulfur amino acid metabolism collectively generate NADPH and glutathione, sustaining the GPX4/TRX antioxidant systems and suppressing ferroptosis. Branched-chain amino acid (BCAA)–mTOR and tryptophan (Trp)–aryl hydrocarbon receptor (AhR) axes couple amino acid sensing to redox signaling, stabilizing the stem-like phenotype. Under TME stress, including hypoxia, acidity, and nutrient competition, exosomes and stromal components reinforce stemness and immune evasion through metabolic and redox crosstalk. Therapeutically, targeting glutamine metabolism (ASCT2/GLS), serine biosynthesis (PHGDH/SHMT), or antioxidant defenses (xCT/GPX4) disrupts reducing power, increases oxidative stress, and enhances the efficacy of chemotherapy, PARP inhibition, and immunotherapy. Biomarkers such as xCT/GPX4 expression, PHGDH levels, Nrf2 activity, and GSH/NADPH ratios may guide patient stratification and response prediction. Overall, understanding the redox–amino acid metabolic network provides a mechanistic basis and translational opportunities for precision metabolic therapies in ovarian cancer.

## 1. Introduction

Ovarian cancer remains one of the most lethal malignancies of the female reproductive system, with late-stage diagnosis, recurrence, and platinum resistance collectively contributing to its poor prognosis [1]. A growing body of evidence highlights that a subpopulation of cancer stem cells (CSCs)—characterized by stemness, differentiation capacity, and therapy resistance—plays a pivotal role in recurrence, metastasis, and treatment failure [2,3]. When signaling that controls normal stem cell differentiation becomes dysregulated, aberrant stemness programs are activated, fueling tumor heterogeneity and malignant progression [4]. Classical signaling axes such as Wnt/β-catenin, Hedgehog, and Notch are central to CSCs stemness and fate determination [5,6,7]. Under therapeutic stress, CSCs enhance drug efflux via ATP-binding cassette (ABC) transporters (ABCB1, ABCC1, ABCG2) and elevate their apoptotic threshold by upregulating anti-apoptotic proteins (Bcl-2, BCL-XL, Mcl-1) while suppressing pro-apoptotic factors (Bax, Bak, Puma), leading to a robust drug-resistant phenotype [8,9].

Metabolic reprogramming and the tumor microenvironment (TME) collectively sustain these adaptive responses. Ovarian tumors and ascites environments are characterized by hypoxia, acidity, nutrient deprivation, and immune suppression [10]. Even in the presence of oxygen, OCSCs preferentially activate glycolysis (the Warburg effect) while their mitochondria continuously generate reactive oxygen species (ROS) through electron leakage in the electron transport chain, forming an amplifying “metabolism–ROS” feedback loop when mitochondrial function is impaired [11]. Rather than maintaining a uniformly low-ROS state, OCSCs dynamically regulate a narrow “ROS window,” where moderate ROS promotes stemness and survival, whereas excessive ROS induces oxidative damage and death [12]. To maintain this balance, OCSCs depend on multilayered antioxidant defenses (e.g., glutathione peroxidase [GPX], thioredoxin systems) and constant NADPH regeneration [13].

Amino acid metabolism serves as a central hub connecting redox control, signal transduction, and biosynthetic demand. Glutamine fuels α-ketoglutarate generation via GLS and transaminases, coupling with the malic enzyme and IDH pathways to produce NADPH. Serine/glycine one-carbon metabolism supplies precursors for nucleotide synthesis and redox homeostasis, while cysteine uptake via system Xc^−^ drives glutathione synthesis, cooperating with GPX4 to suppress ferroptosis [13,14]. Beyond their metabolic roles, amino acids function as signaling cues: leucine and glutamine activate mTORC1, and tryptophan catabolites such as kynurenine modulate stemness and immune evasion through the aryl hydrocarbon receptor (AhR) [15,16]. These networks form a feedback system linking amino acid flux, ROS thresholds, and signaling pathways, directly shaping OCSC responses to chemotherapy, radiotherapy, and targeted agents such as PARP inhibitors.

Current research predominantly approaches the topics of CSCs stemness, amino acid metabolism, redox homeostasis, ferroptosis, or novel therapies separately, yet systematic integration with key factors of the TME—such as hypoxia, acidity, nutrient availability, ascites, and immunometabolism—remains insufficient [17,18,19]. This review proposes a “redox-amino acid-microenvironment” closed-loop perspective, focusing on an integrated framework of ROS threshold regulation and NADPH/GSH supply, while identifying druggable nodes intersecting with pathways like Wnt/mTOR. It consolidates previously fragmented research threads into a dynamic, interactive system rather than merely listing isolated elements. The review explicitly identifies amino acid metabolism as the bridge connecting redox regulation in CSCs with the microenvironment. For instance, tumor cells extensively uptake glutamine and cystine, not only to meet their own antioxidant (GSH synthesis) and energy demands but also depleting these amino acids in the microenvironment. This “nutrient competition” directly suppresses immune cell function, actively shaping a microenvironment conducive to CSCs survival and immune evasion. The discussion extends to combination and timing strategies with chemotherapy, radiotherapy, and immunotherapy. It emphasizes that treatments (e.g., chemo/radiotherapy) generate ROS, while CSCs counteract oxidative stress via their highly responsive amino acid metabolic network (e.g., GSH synthesis). Simultaneously, this metabolic process exacerbates immunosuppression in the TME, weakening the immune system’s ability to clear residual tumor cells, ultimately leading to treatment failure. Novel target combinations for combination therapy are proposed: for example, using amino acid metabolism inhibitors to disrupt the antioxidant foundation and energy supply of CSCs while combining them with chemotherapeutic agents. This dual approach weakens CSCs while alleviating nutrient competition to improve the TME, thereby enhancing the efficacy of subsequent treatments. The concept of “metabolic checkpoints” is redefined. Previously, metabolism was largely viewed as the “logistics department” for cell proliferation, but this review elevates amino acid metabolic pathways (e.g., GSH synthesis) to critical “metabolic checkpoints” that maintain the stability of the entire tumor ecosystem. Targeting these nodes not only starves tumor cells but also dismantles their oxidative stress resistance and reverses immunosuppression, achieving a “three-birds-with-one-stone” effect. This work provides a foundation and direction for deepening the mechanistic understanding of ovarian cancer stem cells (OCSCs) and optimizing therapeutic strategies for ovarian cancer.

## 2. Redox Homeostasis and OCSCs

### 2.1. ROS Threshold and State Plasticity

ROS play dual roles in ovarian cancer as both signaling molecules and damaging agents [12]. Within the TME, the response of OCSCs to ROS is threshold-dependent: levels within a specific range promote cell survival, whereas levels exceeding this threshold induce cell death [12]. Under this framework, OCSCs undergo reversible state: under low ROS/high antioxidant conditions, NADPH and systems like GSH/TRX/PRDX are active, minimizing DNA oxidative damage and maintaining quiescence and stemness, conferring resistance to chemotherapy/radiotherapy [20,21] (Figure 1A, left). Conversely, under high ROS/limited antioxidant conditions, pathways like MAPK/ERK, NOTCH1, and Nrf2 are activated, upregulating ABC transporters to enhance proliferation, migration, and drug resistance [12] (Figure 1A, right). State transitions depend on both stress levels and antioxidant capacity, providing a dynamic basis for recurrence and resistance.

### 2.2. Antioxidant Network and Threshold Crossing

OCSCs maintain redox homeostasis through a hierarchical antioxidant network. Enzymatic systems include SODs converting O_2_·– to H_2_O_2_, with CAT/PRDXs/GPXs further reducing H_2_O_2_ to water (Figure 1B, bottom); GPX4 specifically eliminates lipid peroxides and is a key enzyme determining ferroptosis sensitivity [22,23]. Non-enzymatic systems primarily rely on GSH and TRX, sustained by NADPH regeneration from the pentose phosphate pathway, malate cycle, and IDH1/2 [22] (Figure 1B, right). When ROS production exceeds the network’s capacity or key nodes are pharmacologically/genetically inhibited, “threshold crossing” occurs: apoptosis or ferroptosis is triggered, reducing stemness and enhancing therapy sensitivity [24,25]. Functional studies show that wogonin downregulates PRDX5, elevates ROS, and diminishes stemness [26]; miR-153 suppresses Nrf2, reduces GPX1, and increases ROS, weakening radiation resistance and stemness [27]; apatinib significantly elevates ROS and inhibits CSC-associated phenotypes [28] (Figure 1B, left). These findings support intervention strategies targeting antioxidant bottlenecks (e.g., GPX4, GSH cycling, NADPH supply).

### 2.3. ROS-Mediated Death Pathways and Stemness Suppression

Under high ROS, stress cascades (JNK/p38 MAPK) can activate apoptosis (Figure 1C, top), while ROS suppresses stemness-related signals like Hedgehog, promoting OCSCs differentiation. Differentiated cells exhibit higher apoptotic sensitivity, forming a differentiation-induced sensitization linkage [29,30] (Figure 1C, middle). Additionally, the PSTK-cGAS-STING-ROS positive feedback loop can amplify ROS, inducing lipid peroxidation accumulation and ferroptosis (Figure 1C, bottom). In OCSCs, high ROS levels trigger ferroptosis, reducing colony formation and stemness [31,32]. Ferroptosis is regulated by free iron, membrane polyunsaturated fatty acids, and nodes like xCT/GPX4/NADPH; targeting these nodes in combination with chemotherapy/radiotherapy or PARP inhibition may improve therapeutic responses [22,23,31,32].

## 3. Amino Acid Metabolism and OCSCs

### 3.1. Common Metabolic Reprogramming and Targetable Pathways

CSCs universally exhibit amino acid metabolic reprogramming to meet the demands of rapid proliferation, anti-apoptosis, and stemness maintenance [33,34]. Sulfur-containing amino acids provide precursors for antioxidant systems like glutathione, while glutamine and branched-chain amino acids (BCAAs) contribute to carbon/nitrogen supply and mTOR pathway activation. Tryptophan metabolism influences differentiation and the immune microenvironment via AhR. Targeting key nodes in the “supply transport–critical signaling” axis has shown potential in vitro and in animal models to weaken stemness and enhance chemotherapy and radiotherapy sensitivity [35,36,37,38].

### 3.2. Glutamine and One-Carbon Pathway: Carbon/Nitrogen Supply and NADPH Support

Glutamine provides carbon/nitrogen sources for nucleotides, lipids, and non-essential amino acids, while mTORC1 activation supports proliferation, stemness, and protein synthesis. Various CSCs highly depend on glutamine transporters (e.g., ASCT2, LAT1), and targeting these transporters can reduce stemness markers and tumorigenic capacity [19,39,40,41]. In ovarian cancer models, ASCT2 upregulation often coincides with reduced glycolysis, making the cells more dependent on glutamine flux for energy and biosynthesis. Targeting the “ASCT2-glutamine-alanine” axis significantly inhibits OCSCs stemness and tumor initiation [40]. Mechanistically, glutamine maintains NADPH production and reducing equivalents via transamination and TCA replenishment, reducing oxidative stress and indirectly stabilizing stemness-related pathways [42,43].

### 3.3. Sulfur Metabolism and Ferroptosis Susceptibility: xCT/GPX4 Axis

Sulfur-containing amino acids (methionine, cysteine) regulate ROS levels via the GSH/GPX system, supporting stemness and drug resistance. Methionine supply influences the methyl cycle and epigenetic state, modulating stemness gene expression [35,36]. xCT-mediated cystine uptake and GSH synthesis are critical for GPX4 activity, closely linked to ferroptosis susceptibility. In ovarian cancer, inhibiting xCT or impairing GSH regeneration enhances chemotherapy and radiotherapy sensitivity and synergizes with DNA damage repair inhibitors [39,41,44]. Dietary or metabolic interventions targeting methionine supply are emerging as systemic sensitization strategies, showing potential in combination with metabolic inhibitors and conventional therapies [35,36].

### 3.4. Metabolic-Signaling Coupling: BCAA–mTOR and Trp–AhR

BCAAs, particularly leucine, directly activate mTORC1, driving protein synthesis, proliferation, and resistance. However, mTORC1/2 effects on stemness are context-dependent, requiring careful evaluation with stemness markers and pathway context [45]. Separately, it was discovered that METTL16 promotes the maintenance of stemness precisely through reprogramming BCAA metabolism [46]. In ovarian CSCs, targeting BCAA transporters (e.g., LAT1) and mTOR sensitivity may enable personalized metabolic-signaling strategies. Optimizing transporter and mTOR inhibition timing could enhance OCSC selectivity without increasing toxicity.

Tryptophan metabolites (e.g., FICZ) regulate stemness dose-dependently: low doses suppress stemness genes and promote differentiation, while high levels may promote tumor progression via immunosuppression [38]. ITE induces differentiation and inhibits proliferation via the AhR-Oct4 axis [47]. By analyzing IDO activity and metabolite profiles within the TME, selective AhR modulation may achieve dual benefits of “differentiation + therapy sensitization.” Drug targets should prioritize “transporters (ASCT2/LAT1/xCT)—metabolic enzymes (GLS, AGXT, MDH2)—signaling pathways (mTOR, AhR, NF-κB),” focusing on nodes with in vivo evidence and resistance links. In CSCs models, axes like SLC1A3–NF-κB and MDH2–PDGFRβ show therapeutic promise [48,49,50,51,52] (Table 1).

## 4. Intersection of Redox and Amino Acid Metabolism

### 4.1. Metabolic Hubs: NADPH and GSH Supply Network

CSCs primarily rely on two pillars for redox homeostasis: NADPH and glutathione (GSH). Three amino acid-related pathways provide substrates and reducing equivalents, which determine ROS thresholds and ferroptosis susceptibility: (1) Glutamine axis: ASCT2/SLC1A5 mediates uptake, while GLS and GDH1 convert glutamine to α-KG that enters the TCA cycle, contributing to NADPH production and supporting antioxidant capacity and biosynthetic processes [54]; inhibiting enzymes GLS/GDH1/GOT in this pathway reduces GSH sources, increases ROS, and compromises stemness properties and drug-resistant phenotypes [55,56,57] (Figure 2A). (2) Serine/glycine and one-carbon metabolism: The PHGDH-PSAT1-PSPH pathway and SHMT maintain one-carbon cycling, NADPH generation, and GSH precursor (Gly) supply; simultaneously inhibiting GLS and PHGDH diminishes compensatory pathways and decreases CSCs proportion; SHMT2 deficiency can also trigger cell death [58,59,60,61] (Figure 2B). (3) Sulfur amino acid axis: xCT (SLC7A11) promotes cystine uptake for GSH synthesis, while GPX4 limits lipid peroxidation; xCT/GPX4 upregulation is associated with platinum resistance in ovarian cancer, and their inhibition triggers ferroptosis and reverses tolerance [62,63,64] (Figure 2C). These interconnected pathways create a functional loop of “amino acid supply—NADPH/GSH—ROS control” that directly impacts the maintenance of stemness and therapeutic efficacy.

### 4.2. Metabolic-Signaling Integration: mTOR, Nrf2, and AhR

Amino acid metabolism shapes CSC phenotypes through classical signaling pathways. Glutamine uptake/catabolism often accompanies mTORC1 activation and Nrf2-mediated antioxidant program enhancement, promoting stemness and tolerance; targeting GLS or transporters disrupts this axis and re-sensitizes cells to treatment [41,65,66] (Figure 2A). Serine/glycine metabolism is regulated by ATF4/CEBPB, MYC, and Josephin-2, with PHGDH/SHMT maintaining NADPH and nucleotide synthesis to support stress survival; PHGDH inhibition reduces sphere formation and stemness markers [60,67,68] (Figure 2B). In the sulfur amino acid axis, ROS can upregulate xCT via Keap1/Nrf2, while p53 and mTORC2 suppress xCT activity, thereby modulating ferroptosis thresholds and drug responses [69,70,71,72] (Figure 2C). Additionally, Myc participates in tryptophan-kynurenine (Kyn) pathway regulation; the Kyn pathway influences NAD+ metabolism and, under specific conditions, reduces ROS to activate Nrf2 or AhR, maintaining xCT activity, affecting stemness, and changing cell death mechanisms [73,74,75] (Figure 2D).

However, these factors do not regulate cellular states through simple “on/off” mechanisms. In the regulation of CSCs, they often exhibit multifaceted roles. Firstly, mTORC1 serves as a critical serine/threonine kinase complex and acts as a central hub for cellular nutrient and energy sensing [76]. Its activity is precisely regulated by redox status: elevated ROS levels can sustain mTORC1 activation through pathways such as AMPK and HIF-1α, thereby remodeling cellular metabolism. Conversely, mTORC1 activation enhances anabolic processes, promotes the reinforcement of antioxidant barriers, and enables cells to adapt to and exploit this high-ROS environment, ultimately establishing a positive feedback loop that drives tumorigenesis [77]. In the context of stemness regulation, mTORC1 functions as a core regulator. It finely coordinates biosynthetic and metabolic programs to maintain the stemness of stem cells while also directing their transition toward differentiation, thereby dynamically balancing stem cell fate [78]. Disruption of this delicate balance leads to dysregulation of stemness homeostasis. Overall, mTORC1 plays a pivotal role in determining stem cell fate by integrating metabolic signals and oxidative stress responses.

Furthermore, AhR, as a ligand-activated transcription factor, exhibits remarkable context-dependent functionality. This characteristic is particularly evident in redox regulation: on one hand, AhR can be activated by pro-oxidant ligands to induce the expression of cytochrome P450 enzymes (such as CYP1, CYP2, and CYP3), thereby promoting ROS production and exacerbating oxidative stress [79]; on the other hand, it can also bind to antioxidant ligands and directly regulate the core transcription factor Nrf2 to enhance the expression of antioxidant enzymes, thereby strengthening the cellular antioxidant defense system [80]. This demonstrates the unique dual role of AhR in redox homeostasis, with the ultimate outcome depending on the nature of the ligands. Its role in stemness maintenance is equally complex: although AhR activation or overexpression has been shown to mediate the growth and maintenance of CSCs in various gynecological cancers [81], it may exert opposite inhibitory effects under specific TME contexts, suppressing CSCs formation [82]. These findings indicate that the net effect of AhR is primarily determined by the specific cellular context and microenvironmental signals.

## 5. Tumor Microenvironment–Driven Metabolic Remodeling in OCSCs

### 5.1. Hypoxia and Acidosis: Redox Adaptation and Stemness Preservation

Hypoxia and acidity are core features of the ovarian cancer TME, closely associated with CSCs maintenance, invasion, and drug resistance. Hypoxia activates hypoxia-inducible factor (HIF) signaling to inhibit mitochondrial oxygen consumption, enhance glycolysis, and remodel antioxidant programs, thereby reducing oxidative stress, maintaining stemness and quiescence, and enhancing radio/chemoresistance and metastatic potential [83,84,85,86]. Under nutrient limitation, HIF-1α upregulates serine and glycine synthesis pathways (e.g., PHGDH/PSAT1/PSPH) via the ROS–AMPK axis, supplying NADPH and GSH precursors to bolster antioxidant capacity, spheroid formation, and stemness [87]. In ovarian cancer models, hypoxia induces HIF-1α to upregulate solute carrier family 2 member 12 (SLC2A12), engaging glutathione metabolism, inhibiting ferroptosis, and promoting tumor progression and drug resistance [88]. Conversely, HIF deficiency increases ROS and activates p53/p16(Ink4a), inducing CSCs senescence or death, underscoring its pivotal role in stemness maintenance [84] (Figure 3A, top).

Lactate accumulation drives acidity, promoting epithelial–mesenchymal transition (EMT), upregulating stemness markers like SOX2, increasing clonogenicity, and associating with drug resistance [89,90,91]. Monocarboxylate transporters 1/4 (MCT1/4)-mediated lactate transmembrane transport facilitates invasive remodeling [92]; acidity also lifts SOX2 transcriptional repression by reducing peroxisome proliferator-activated receptor delta (PPARD) and inhibiting vitamin D receptor (VDR) activity, bolstering stemness and chemoresistance, while VDR reconstitution reverses this phenotype [93]. Collectively, hypoxia and acidity underpin ovarian cancer CSCs adaptive survival through the “metabolic reprogramming—enhanced antioxidant capacity—phenotypic remodeling” axis (Figure 3A, bottom).

### 5.2. Exosome-Mediated Metabolic Signaling and Immune Modulation

Exosomes transport miRNAs and proteins between ovarian cancer cells and TME components, driving bidirectional metabolic and signaling remodeling to promote stemness, angiogenesis, and drug resistance [94]. Key mechanisms include: tumor-derived exosomes carrying miRNA-155 upregulate superoxide dismutase 2/catalase (SOD2/CAT), reducing ROS to bolster survival and drug resistance [95]; ascites-derived exosomal miRNA-6780b-5p drives EMT and metastatic spread [96]; exosomes containing prokineticin receptor 1 (PKR1) or miRNA-130a fuel angiogenesis and drug resistance [97,98]; CSC-derived exosomes carrying miRNA-210 activate mechanistic target of rapamycin (mTOR), promoting drug resistance [99]. From the TME, fibroblast-derived exosomal miRNA-92a-3p induces drug resistance and tumor growth via Wnt/β-catenin signaling [100]; HJURP-containing exosomes enhance glutamine metabolism and TCA flux, inducing drug resistance [101] (Figure 3B, top). Exosomes carrying CD39/CD73 drive macrophage polarization to M2 via adenosine A2B receptor (A2BR), secreting angiogenic factors to promote tumor progression [102]; macrophage-derived exosomal miRNA-3679-5p inhibits neural precursor cell-expressed developmentally downregulated 4-like (NEDD4L), enhancing c-Myc oncogenic signaling and fueling tumor growth and cisplatin resistance [103]. Under hypoxia, IL-6 and miRNA-155-3p in tumor exosomes synergistically amplify autophagy and M2 polarization via STAT3, driving tumorigenesis [104]; ovarian cancer exosomal miRNA-940 also induces M2 polarization, fueling progression [105] (Figure 3B, bottom). Exosomes thus promote CSCs dominance via three synergistic pathways: “enhanced antioxidant capacity—improved nutrient metabolism—immune remodeling.”

### 5.3. Nutrient Competition and Metabolic-Immunological Crosstalk

In the TME, tumor and immune cells compete for critical amino acids, including glutamine, serine, arginine, and tryptophan, altering immune cell metabolism and impairing anti-tumor functions [106,107]. Effector T cells exhibit heightened dependence on glutamine, serine, and arginine (Figure 3C, top); glutaminase (GLS) deficiency inhibits Th17 differentiation and remodels Th1/CD8+ T cell effector functions [107,108] (Figure 3C, right). B cells rely more on glutamine metabolism under hypoxia [109] (Figure 3C, middle). Macrophage polarization coincides with metabolic rewiring: M1 macrophages enhance glutaminolysis and accumulate succinate to drive HIF-1α-mediated inflammation [110], while M2 macrophages depend on glutamine to fuel biosynthesis via TCA cycle intermediates [111] (Figure 3C, bottom). Under resource limitation, tumor cells deploy multiple mechanisms to suppress immunity: fumarate hydratase (FH) and succinate dehydrogenase (SDH) mutations lead to oncometabolite accumulation (e.g., fumarate, succinate), depleting α-KG and compromising Th17 differentiation [112] (Figure 3D, left); tumor cells compete with conventional type 1 dendritic cells (cDC1) for SLC38A2-mediated glutamine uptake, compromising anti-tumor immunity [113] (Figure 3D, right); arginase-1 (Arg-1)-mediated arginine depletion paralyzes T cells [114]; indoleamine 2,3-dioxygenase (IDO) catabolizes tryptophan, suppressing effector T/natural killer (NK) cells and triggering apoptosis [115] (Figure 3D, middle). Therefore, interventions targeting metabolic competition (e.g., context-specific inhibition of nodes like GLS/PHGDH/xCT/IDO/Arg-1), paired with coordinated immune-metabolic enhancement, could reverse immune suppression and eradicate CSCs.

## 6. Translational Strategies and Proximal Evidence

Based on the aforementioned nodal intersections, actionable translational strategies can be formulated as follows: (1) Disrupt redox balance to break down defense lines: Directly target GSH, xCT, or GPX4 to alter ROS levels. Enhance lipid peroxidation and induce ferroptosis—particularly effective for high Gln/Glu or platinum-resistant phenotypes [62,63,64]. (2) Regulate amino acid metabolism to weaken supply pathways: Inhibit ASCT2/GLS/GDH1 or PHGDH/SHMT to indirectly compromise defensive barriers, thereby reducing stemness and improving therapeutic sensitivity [59,60,61]. For example, dual inhibition of GLS and PHGDH has been shown to effectively diminish CSCs and recurrence risk across multiple models [60]. (3) Synergistic network strategy: Combine amino acid metabolism modulation with redox balance regulation to simultaneously suppress nutrient signaling and antioxidant compensation [66,70], enabling a combinatorial approach for CSC-targeted therapy.

### 6.1. Increasing ROS Load and Disrupting Antioxidant Defenses

CSCs typically maintain low ROS levels to preserve stemness and tolerance. Increasing ROS load or weakening the antioxidant system can enhance their sensitivity to radiotherapy, chemotherapy, and ferroptosis (iron-dependent cell death). Representative strategies include: using BSO to promote ROS production and improve radiosensitivity [116]; Resveratrol increases ROS and activates the AMPK–TSC–mTOR pathway to enhance temozolomide efficacy [117]; arsenic trioxide inhibits CSCs via ROS-dependent mechanisms [118]; small-molecule ferroptosis inducers and Salinomycin trigger ferroptosis by elevating ROS, which is more effective against CSCs [119,120]; PEITC eliminates drug-resistant CSCs by depleting GSH and disrupting reducing capacity [121]. Moreover, downregulation of GPX4 activity promotes ROS generation and stimulates lipid peroxidation [64], ultimately inducing ferroptosis and thereby counteracting chemotherapy resistance in CSCs [122].Combination regimens such as erastin + doxorubicin (DOX) can deplete GSH, inhibit drug efflux, and promote ROS accumulation, achieving synergistic killing [30]. Some nanoparticle-based systems (e.g., P@Ce6/PTX, ZnPP@FQOS) exacerbate oxidative stress by simultaneously generating ROS and weakening the NADPH/GSH defense system, with low systemic toxicity [123,124].

### 6.2. Intervening in Amino Acid Dependence and Signaling Coupling

Ovarian cancer and OCSCs exhibit dependence on metabolic pathways such as glutamine and serine/glycine. Restricting glutamine or inhibiting downstream mTORC1/S6K signaling can reduce stemness and reverse paclitaxel resistance [125]. Targeting GLS, a key enzyme in glutamine metabolism, through genetic knockdown or pharmacological inhibition (e.g., using CB-839) can induce glutamine deprivation and ROS accumulation in cancer cells, subsequently triggering ferroptosis in ovarian cancer cells [55]. Inhibition of GLS also promotes the differentiation and effector functions of CD4^+^ Th1 and CD8^+^ CTL cells [108], while enhancing tumor sensitivity to radiotherapy [66], demonstrating that targeting GLS represents a potential strategy for ovarian cancer treatment. Furthermore, studies indicate that simultaneous inhibition of GLS and PHGDH, a critical enzyme in the serine/glycine metabolic pathway, can synergistically eliminate cancer stem cells [60]. Aurora-A is associated with serine/threonine metabolism and stemness upregulation [126]; KDM5B-mediated lysine demethylation promotes differentiation and compromises stemness [127]. Near-term clinical targets prioritize the “glutamine–mTORC1” and “serine/glycine–GSH/one-carbon metabolism (folate cycle)” axes, while other molecular nodes serve as background support and biomarkers for patient stratification.

### 6.3. Synergistic Targeting of Redox-Amino Acid Networks: Combinations and Timing

Combining metabolic inhibition with oxidative stress amplification can simultaneously deprive cells of metabolic precursors and disrupt the NADPH/GSH defense system, thereby enhancing the killing of CSCs and the tumor bulk. Examples include: Combined administration of methionine/cysteine inhibitors with a GPX4 inhibitor (RSL3) exacerbates lipid peroxidation and ferroptosis in tumor cells, thereby suppressing stemness [128]. NCT 503@Cu HMPB inhibiting PHGDH to block serine synthesis and GSH production, inducing oxidative imbalance; TiO_2_ Au@DON reprogramming glutamine metabolism with DON to weaken NADPH supply, while TiO_2_ Au promotes ROS elevation and induces immunogenic cell death (ICD), synergistically reshaping the immunosuppressive microenvironment. Combination designs should prioritize complementary pairings of “metabolic inhibition (PHGDH/GLS/xCT) + ROS amplification (photosensitization/metal catalysis/DOX)” and incorporate biomarkers such as xCT, PHGDH, GLS expression, and GSH/NADPH ratios for patient stratification and efficacy prediction [129,130] (Table 2).

## 7. Conclusions and Outlook

The metabolic and redox characteristics of OCSCs are gradually forming a clear framework: OCSCs typically maintain low and controlled ROS levels, buffered by antioxidant systems such as NADPH/GSH and TRX/PRDX [22], and are tightly coupled with glutamine metabolism, serine/glycine and one-carbon pathways, BCAAs, and the tryptophan–kynurenine (Kyn) axis [34]. These metabolic networks are amplified or stabilized by signaling pathways such as mTORC1/ATF4, Nrf2, AhR, and NOTCH [15,16] under TME conditions such as hypoxia, acidity, and nutrient competition, shaping a homeostatic niche that supports stemness maintenance, drug tolerance, and immune evasion. This synthetic perspective links “redox balance–amino acid metabolism–signaling pathways–TME” into a closed loop explaining drug resistance and recurrence, providing a roadmap for identifying druggable cross-nodes.

Based on this framework, several strategies with translational potential have begun to take shape. Supply-side restrictions can deplete precursors and reduce equivalents by inhibiting key nodes such as PHGDH/SHMT (one-carbon pathway), GLS (glutamine), xCT (cystine/glutathione axis), and IDO (tryptophan metabolism) [60]. Stress-side amplification can disrupt antioxidant buffering and membrane protection by inhibiting GPX4 or TRX, suppressing the Nrf2 pathway, and promoting lipid peroxidation and ferroptosis [64]. Meanwhile, combining chemotherapy/radiotherapy, PARP inhibition, immunotherapy, and other treatments with the above metabolic–redox regulation, either sequentially or simultaneously, is expected to achieve dual targeting of OCSCs and their niche through “weakening supply + amplifying stress,” thereby improving clearance efficiency and delaying recurrence.

When translating therapeutic strategies targeting redox and amino acid metabolism to the clinic, it is essential to systematically evaluate their inherent limitations while focusing on their tumor-killing effects. First, although direct induction of ferroptosis is highly effective, existing drugs face challenges with systemic toxicity, and developing tissue-specific or activatable prodrugs remains a major hurdle. Second, small-molecule inhibitors targeting GLS or xCT often demonstrate limited efficacy as monotherapies, necessitating exploration of synergistic approaches with radiotherapy, chemotherapy, or immunotherapy. Third, the high plasticity of metabolic networks allows cancer cells to develop compensatory mechanisms that undermine single-target interventions. These include microenvironmental adaptations such as enhanced amino acid transport in response to GLS inhibition, increased glutamine dependency due to reduced glucose utilization, and adaptive activation of antioxidant defense systems.

To facilitate clinical translation, a three-pronged actionable pathway is proposed: First, establish a stratification system centered on metabolic–redox status, combined with tissue and liquid biopsy markers (e.g., xCT/GPX4 and Nrf2 activity, ratios of reduced/oxidized glutathione (GSH/GSSG) and NADPH/NADP+, lipid peroxidation, and ferroptosis indicators) and functional imaging, to enable patient selection and dose/sequence optimization [63]. Second, evaluate combination strategies of “restricting supply + amplifying stress” in organoids, PDX, and composite TME models to clarify selective killing of OCSCs and non-stem cell populations, toxicity thresholds, and tolerance, and to determine optimal dosing sequences [123]. Third, conduct prospective trials with unified efficacy and biomarker endpoints (e.g., recurrence-free survival, dynamic changes in OCSC-related markers) to enhance the comparability and reproducibility of evidence.

In summary, targeting the cross-nodes of redox homeostasis and amino acid metabolism, combined with mechanism-driven combination therapies and biomarker stratification, offers a promising approach to addressing drug resistance and recurrence challenges in ovarian cancer. With the advancement of cross-scale models and clinical validation, this integrated strategy is expected to translate into more stable and reproducible clinical benefits, providing a practical intervention framework for the treatment of advanced drug-resistant and recurrent ovarian cancer.

## Figures and Tables

**Figure 1 antioxidants-14-01413-f001:**
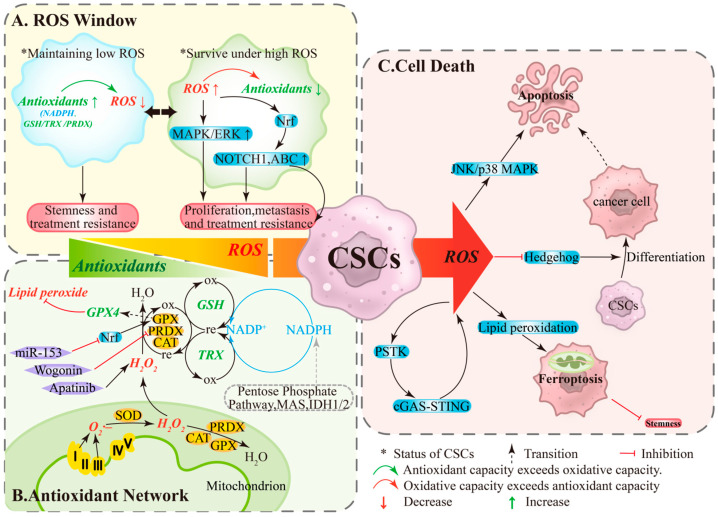
**The ROS Window, Antioxidant Network, and Cell Death Pathways in Ovarian Cancer Stem Cells**. (**A**) **ROS Window:** Low ROS levels and high antioxidant activity help maintain stemness and treatment tolerance, whereas high ROS levels and limited antioxidant capacity can paradoxically enhance proliferation, migration, and drug resistance. (**B**) **Antioxidant Network:** Mitochondrial ROS are reduced to water by CAT/PRDXs/GPXs, a process supported by non-enzymatic systems (GSH and TRX) and dependent on the continuous regeneration of NADPH. (**C**) **Cell Death:** When ROS production exceeds the processing capacity of this network, a “threshold crossing” occurs, impairing stemness through multiple pathways. ABC, ATP-binding cassette transporters; CAT, catalase; GPX, glutathione peroxidase; GSH, glutathione; IDH, Isocitrate Dehydrogenase; MAPK/ERK, mitogen-activated protein kinase/extracellular signal-regulated kinase; MAS, malate-aspartate shuttle; Nrf2, nuclear factor erythroid 2-related factor 2; ox, oxidation; PRDX, Peroxiredoxin; re, reduction; TRX, thioredoxin.

**Figure 2 antioxidants-14-01413-f002:**
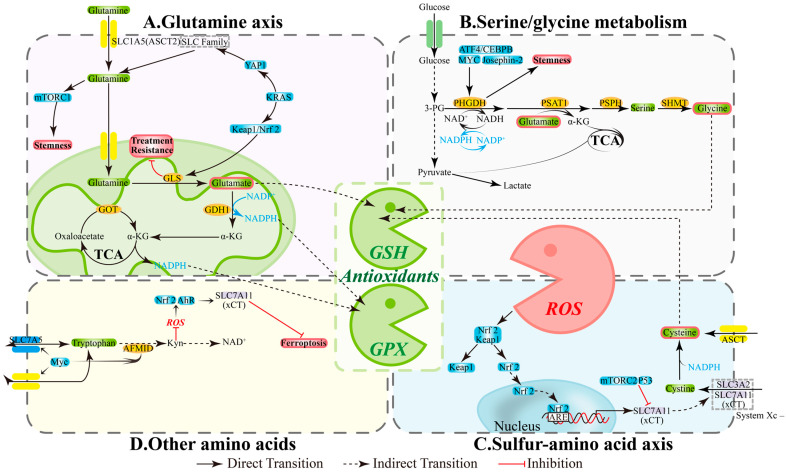
**The intersecting hub of redox balance and amino acid metabolism**. (**A**) **Glutamine axis:** Central Redox Coordinator. (**B**) **Serine/glycine metabolism:** Antioxidant Arsenal Factory. (**C**) **Sulfur-amino acid axis:** Reactive Core for Redox Homeostasis. (**D**) **Other amino acids:** Synergistic Redox Network Guardians. AFMID, enzyme arylformamidase; AhR, aryl hydrocarbon receptor; Akt, Ak strain transforming; ARE, antioxidant response element; ASCT2, alanine-serine-cysteine transporter 2; ATF4, activating transcription factor 4; GDH1, glutamate dehydrogenase 1; GLS, glutaminase; GOT, glutamic-oxaloacetic transaminase; GPX, Glutathione Peroxidase; GSH, glutathione; IDO1/2, Indoleamine 2,3-dioxygenase 1/2; Keap1, kelch-like ECH-associated protein 1; KRAS, Kirsten RAt Sarcoma virus oncogene; mTOR, mechanistic target of rapamycin; Myc, Myelocytomatosis oncogene; NRF2, nuclear factor erythroid 2-related factor 2; PHGDH, phosphoglycerate dehydrogenase; PI3K, phosphatidylinositol 3-kinase; PSAT1, phosphoserine aminotransferase 1; PSPH, phosphoserine phosphatase; ROS, reactive oxygen species; SHMT, serine hydroxymethyltransferase; SLC1A5, solute-linked carrier family A1 member 5; YAP1, Yes-associated protein 1; α-KG, α-ketoglutarate.

**Figure 3 antioxidants-14-01413-f003:**
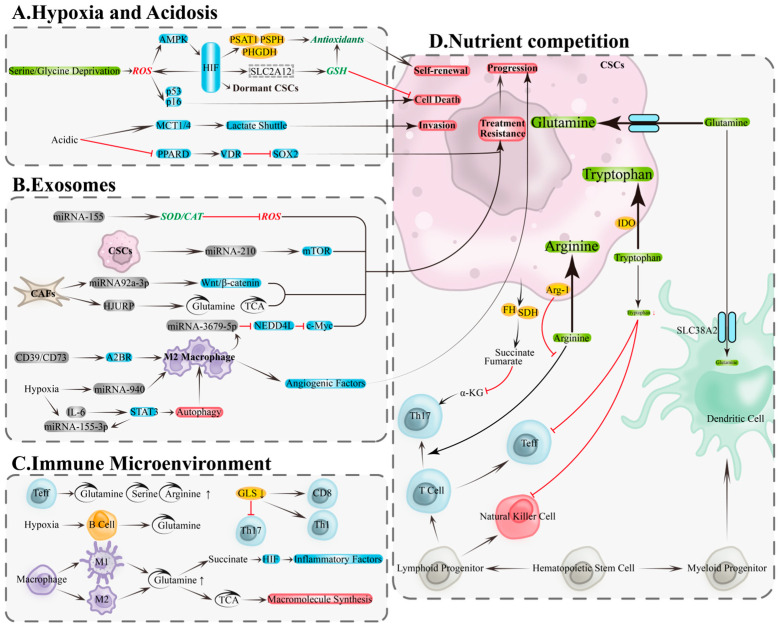
**Metabolic and immune remodeling in the tumor microenvironment: hypoxia/acidity, exosomes, and nutrient competition.** (**A**) Metabolic remodeling of CSCs in hypoxic and acidic microenvironments is a key foundation for the adaptive survival mechanisms of OCSCs. (**B**) Exosomes drive bidirectional regulation of metabolism and signaling through their functional molecules, including proteins, lipids, and nucleic acids (DNA and RNA). (**C**) Metabolic remodeling within immune cells. (**D**) Nutrient competition between tumor and immune cells. A2BR, adenosine A2B receptor; AMPK, AMP-activated protein kinase; Arg-1, arginase-1; CAT, catalase; EMT: epithelial–mesenchymal transition; FH, fumarate hydratase; GLS, glutaminase; GSH, glutathione; HIF, hypoxia-Inducible Factor; HJURP, Holliday junction recognition protein; IDO, indolamine-2,3-dioxygenase; MCT1/4, monocarboxylate transporters 1/4; mTOR, mechanistic target of rapamycin; Myc, myelocytomatosis oncogene; NEDD4L, neural precursor cell expressed developmentally down-regulated 4-Like; PHGDH, phosphoglycerate dehydrogenase; PKR1, prokineticin receptor 1; PPARD, peroxisome proliferator-activated receptor delta; PSAT1, phosphoserine aminotransferase 1; PSPH, phosphoserine phosphatase; SDH, succinate dehydrogenase; SOD2, superoxide dismutase 2; SOX2, SRY-Box transcription factor 2; STAT3, signal transducer and activator of transcription 3; TCA, tricarboxylic acid cycle; VDR, vitamin D receptor; α-KG, α-ketoglutarate. Black arrows, direct Transition; red arrows, inhibition.

**Table 1 antioxidants-14-01413-t001:** Key amino acids and related molecular targets involved in CSCs phenotypes.

Identify	Target	Function	References
Methionine	PGC-1α/PPAR-α	Restriction inhibits CSCs metastasis and recurrence.	[35,36]
Tryptophan	AhR/Oct 4	FICZ (derivative): Low doses inhibit stemness and promote differentiation; high levels facilitatetumor progression via immunosuppression. ITE (metabolite): Induces CSCs differentiation and inhibits proliferation.	[37,38,47]
BCAAs	mTORC1/2	Leucine controls tumor proliferation and drug resistance. BCAAs influence stemness and chemotherapy sensitivity via mTORC1/2.	[45,53]
Glutamine	mTORC1, ASCT2	Promotes stem cell phenotype. Deprivation eliminates CSCs.	[19,39,40,41,44]
Alanine	Akt, AGXT/SOX2-OCT4	OBA-RT (novel compound) inhibits Akt pathway, weakens stemness, and induces apoptosis. AGXT regulates stem cell markers (SOX2, OCT4) to eliminate CSCs.	[48,49]
Glutamate	CD133/NF-κB/SLC1A3	Inhibiting glutamate levels suppresses CSCs activity.	[50]
Aspartate	MDH2/α-KG/m6A/PDGFRβ	Promotes CSCs proliferation and stemness via the MDH2-PDGFRβ axis.	[51,52]

Abbreviations: α-KG: alpha-ketoglutarate; AhR: aryl hydrocarbon receptor; AGT: alanine–glyoxylate aminotransferase; AGXT, Alanine–glyoxylate and serine–pyruvate aminotransferase; BCAAs: Branched-chain amino acids; FICZ: 6-formylindolo [3,2-b]carbazole; ITE: 2-(1′ H-indole-3′-carbonyl)-thiazole-4-carboxylic acid methylester; Leu: leucine; m6A: N6-methyladenosine; MDH2: malate dehydrogenase 2; mTORC1/2: mammalian target of rapamycin complex 1/2; OBA-RT: 5-O-(N-Boc-l-alanine)-renieramycin T; PDGFRβ: platelet-derived growth factor receptor-β; Trp: Tryptophan.

**Table 2 antioxidants-14-01413-t002:** Combination strategies and application scenarios targeting the redox–amino acid network.

Translation Strategy	Therapeutic Agents	Target Hubs	Effect	References
Disrupt redox homeostasis	BSO	ROS	Radiotherapy-sensitive	[116]
	Resveratrol		Apoptosis/ferroptosis	[117]
	Arsenic Trioxide		Inhibits CSCs	[118]
	Small-molecule ferroptosis inducers		Ferroptosis	[119]
	Salinomycin		Ferroptosis	[120]
	PEITC	GSH	Eliminates CSCs	[121]
	GPX4 inhibitors	GPX4	Ferroptosis/chemotherapy-sensitive	[122]
	Erastin + Doxorubicin	ROS/GSH	Synergistic killing CSCs	[30]
	P@Ce6/PTX		Synergistic killing CSCs	[123]
	ZnPP@FQOS		Synergistic killing CSCs	[124]
Regulate amino acid metabolism	The selective glutamine transporter ASCT2 inhibitor	Glutamine	Inhibits CSCs	[125]
	CB-839		Ferroptosis/Radiotherapy-sensitive/immunoenhancement	[55,66,108]
	GLS inhibitor + PHGDH inhibitor	Glutamine/Serine	Eliminates CSCs	[60]
	Aurora-A inhibitor	Serine/threonine	Inhibits proliferation	[126]
	KDM5 inhibitors	Lysine	Promotes differentiation	[127]
Synergistically target the redox–amino acid network	Methionine inhibitor/Cysteine inhibitor + GPX4 inhibitors (RSL3)	GSH/GPX4	Inhibits CSCs	[128]
	NCT 503@Cu HMPB	Serine/GSH	Inhibits CSCs	[129]
	TiO_2_ Au@DON	Glutamine/ROS	Inhibits CSCs	[130]

Abbreviations: Aurora-A, Aurora kinase A; BSO, Buthionine sulfoximine; DON, 6-Diazo-5-oxo-l-norleucine; DOX, doxorubicin; GSH, glutathione; KDM5B, Lysine Demethylase 5B; PEITC, Phenethyl Isothiocyanate; PHGDH, phosphoglycerate dehydrogenase; ROS, reactive oxygen species.

## Data Availability

No new data were created or analyzed in this study. Data sharing is not applicable to this article.

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
