# Peer review of "Redox–Amino Acid Metabolic Crosstalk in Ovarian Cancer Stem Cells: Integrating Metabolic Reprogramming, Signaling, and the Tumor Microenvironment"

_antioxidants, 2025, doi:10.3390/antiox14121413_

Round 1

Reviewer 1 Report

This is a very thorough review of the subject but quite difficult.  This is not surprising given the complexity of the problem and why the cancer is so difficult to treat.  Any discussion on drug targets for future treatments is to be welcomed. 

Figures are very complex and take a lot of time to understand the schemes. Table 1 (as shown in the draft provide) is not clear. This may be because the columns for function are centralised with bullet points rather than to the left.

This is a review article. The statements on Author contributions and Conflicts of interest, do not seem appropriate. It is not original research and did not require a study design. 

Non except for Contributions and Conflicts of Interest

Reviewer 2 Report

In this manuscript, Liu et al review how redox-ammino acid metabolism is utilized by ovarian cancer stem cells as a component of their resistance to chemotherapy. They cover various aspects, including reactive oxygen species (ROS ) and anti-oxidant pathways, the roles of amino acid metabolism (including the specific roles of individual a.a.’s),interactions between these, the role of the microenvironment and potential areas for intervention.

Overall, this constitutes an important and timely review of a potentially key vulnerability in an as-of-yet essentially targetable driver of ovarian cancer resistance and recurrence. While well-organized overall, I would suggest that the interventional sections (4.3 and 6.1-6.3) be combined into a single whole.

I have no further comments.

Reviewer 3 Report

Liu et al. submitted a review titled “Redox–Amino Acid Metabolic Crosstalk in Ovarian Cancer Stem Cells: Integrating Metabolic Reprogramming, Signaling, and the Tumor Microenvironment.” The manuscript is comprehensive and well-structured, but several critical points should be addressed before it is suitable for publication.

  • The review mainly summarizes existing literature and lacks a distinctive conceptual advance. Please clarify how the proposed “redox–amino acid–TME framework” offers new mechanistic or therapeutic insight beyond previous models.

  • Critical discussion is limited. Include balanced evaluation of controversial or conflicting findings (e.g., dual effects of mTORC1, AhR, and Nrf2 on stemness and ferroptosis).

  • The text is dense and often repetitive. Careful language editing is needed to improve clarity, shorten sentences, and maintain focus in each paragraph.

  • Figures are detailed but not consistently referenced. Each figure should be clearly cited in the main text, and figure legends should be simplified to highlight the main message only.

  • Replace secondary review citations with primary experimental references where possible, particularly for mechanistic pathways such as ferroptosis, NADPH regeneration, and amino acid signaling.

  • The translational section should be condensed and reorganized to emphasize actionable therapeutic targets (GLS, PHGDH, xCT, GPX4, IDO) and discuss their limitations or potential compensatory mechanisms.

  • Use terminology consistently (e.g., “stemness,” “plasticity,” “self-renewal”) and ensure abbreviations are uniformly defined.

  • A concise schematic or summary table highlighting druggable cross-nodes or clinical biomarkers would enhance translational value.

With these revisions, the manuscript could become a valuable and authoritative reference in redox and metabolic oncology.

Liu et al. submitted a review titled “Redox–Amino Acid Metabolic Crosstalk in Ovarian Cancer Stem Cells: Integrating Metabolic Reprogramming, Signaling, and the Tumor Microenvironment.” The manuscript is comprehensive and well-structured, but several critical points should be addressed before it is suitable for publication.

  • The review mainly summarizes existing literature and lacks a distinctive conceptual advance. Please clarify how the proposed “redox–amino acid–TME framework” offers new mechanistic or therapeutic insight beyond previous models.

  • Critical discussion is limited. Include balanced evaluation of controversial or conflicting findings (e.g., dual effects of mTORC1, AhR, and Nrf2 on stemness and ferroptosis).

  • The text is dense and often repetitive. Careful language editing is needed to improve clarity, shorten sentences, and maintain focus in each paragraph.

  • Figures are detailed but not consistently referenced. Each figure should be clearly cited in the main text, and figure legends should be simplified to highlight the main message only.

  • Replace secondary review citations with primary experimental references where possible, particularly for mechanistic pathways such as ferroptosis, NADPH regeneration, and amino acid signaling.

  • The translational section should be condensed and reorganized to emphasize actionable therapeutic targets (GLS, PHGDH, xCT, GPX4, IDO) and discuss their limitations or potential compensatory mechanisms.

  • Use terminology consistently (e.g., “stemness,” “plasticity,” “self-renewal”) and ensure abbreviations are uniformly defined.

  • A concise schematic or summary table highlighting druggable cross-nodes or clinical biomarkers would enhance translational value.

With these revisions, the manuscript could become a valuable and authoritative reference in redox and metabolic oncology.

Round 2

Reviewer 1 Report

Version 2 is nicely improved including shortening of the Figure legends and increased text where necessary e.g. on AHR.  I would encourage the authors or journal handlers to make Tables 1 and 2 more readable. They look messy.

Other than suggestion of tables, none.

Reviewer 2 Report

The authors have addressed my comments to my satisfaction.

No further comments.

Reviewer 3 Report

The authors have done substantial revision and responded thoroughly to all comments. They expanded the conceptual justification, improved figure clarity, strengthened the dual-role discussion for mTORC1/AhR, reorganized the translational sections, and standardized terminology. The revisions satisfactorily address the previous concerns. I recommend acceptance.

The authors have done substantial revision and responded thoroughly to all comments. They expanded the conceptual justification, improved figure clarity, strengthened the dual-role discussion for mTORC1/AhR, reorganized the translational sections, and standardized terminology. The revisions satisfactorily address the previous concerns. I recommend acceptance.